# Impact of Lifestyle Variables on Oral Diseases and Oral Health-Related Quality of Life in Children of Milan (Italy)

**DOI:** 10.3390/ijerph17186612

**Published:** 2020-09-11

**Authors:** Daniela Carmagnola, Gaia Pellegrini, Matteo Malvezzi, Elena Canciani, Dolaji Henin, Claudia Dellavia

**Affiliations:** 1Department of Biomedical, Surgical and Dental Sciences, Università degli Studi di Milano, 20133 Milano, Italy; daniela.carmagnola@gmail.com (D.C.); elena.canciani@unimi.it (E.C.); dolaji.henin@hotmail.it (D.H.); claudia.dellavia@unimi.it (C.D.); 2Department of Clinical Sciences and Community Health, Università degli Studi di Milano, 20100 Milano, Italy; matteo.malvezzi@unimi.it

**Keywords:** children, Italy, lifestyle, oral disease, oral health-related quality of life, feeding habits, oral hygiene, socio-economic status

## Abstract

A large part of the Italian population doesn’t receive adequate information and support on how to maintain oral health. In this observational, cross-sectional, pilot study, we investigated how some lifestyle-related variables affect oral diseases and oral health-related quality of life (OHRQoL) of children attending public-school summer services in Milan. A survey that included questions on children’s oral disease, OHRQoL and lifestyle-related factors (feeding habits, oral hygiene protective behaviors, dental coaching and socio-economic and educational status), was administered to the children’s caregivers. Data from 296 surveys were analyzed to assess the protective/negative effect of each variable on oral disease and OHRQoL. With respect to disease, the “never” consumption of fruit juice, the use of fluoride toothpaste, higher educational qualification and ISEE (equivalent family income) of those who filled out the form, resulted protective factors. Regarding OHRQoL, the “never” assumption/use of tea bottle, sugared pacifier and fruit juice as well as the use of fluoride toothpaste, a higher educational qualification and ISEE of those who filled out the form, resulted to have protective effects. In conclusion, protective behaviors and socio-economic status affect oral disease and OHRQoL in children of Milan.

## 1. Introduction

Oral diseases such as caries and periodontal inflammation are known to have a multifactorial etiology [1]. Some of their associated risk factors, like poor oral hygiene and elevated sugar intake can be modified, as well as known protective factors, such as the use of fluoride to prevent caries development [2].

Oral diseases are widely spread worldwide and untreated caries in permanent teeth are considered the most prevalent condition among those evaluated in the Global Burden of Disease 2010 study [3]. It is estimated that 60 to 90% of children and adults suffer from caries [4,5] corresponding to approximately 2.4 [6] to 3.5 billion [3] people with untreated oral diseases, i.e., more than 35% of the global population. Concerning deciduous teeth, caries was reported to be the 10th most prevalent condition, affecting 9% of the population, or 621 million children worldwide [3]. Untreated oral diseases can lead to functional, aesthetic and psychological problems resulting in a poor quality of life [7,8]. Quality of life indicators investigate very crucial issues: children with poor oral health limit their activity days, including school days, up to 12 times more than healthy children [8] and it has been estimated that oral diseases, like caries in both the primary and permanent dentition, can be responsible for the loss of over 50 million hours from schools annually worldwide [9,10].

The prevention of dental caries in children and adolescents is also generally regarded as a priority for dental services since its treatment requires time and economic resources that are not available for all individuals and can weigh heavily on healthcare systems. According to the WHO [11], dental caries is the fourth most expensive chronic disease to treat on a population basis. In general, the prevalence of oral diseases and dental caries vary considerably between and within different countries, with children in the lower socioeconomic status groups having higher caries levels than the wealthier ones and such difference can be even more evident in high-income countries. Furthermore, the greatest burden of oral diseases affects the less advantaged fragments of the population: despite the fact that in high-income countries with preventive oral care programs, the prevalence of both dental caries in children and tooth loss among adults has decreased, even in the Nordic countries, social class differences concerning oral health and care access can be observed [11]. It is known that the social determinants of health such as income, education, and behaviors, beyond aspects like use of fluorides, diet, and access to preventive and restorative care services, play the most important role in caries initiation and progression. On a local level, studies have proven how dental caries often tend to cluster in more disadvantaged areas even within a city [12,13].

For all the above cited reasons, the WHO recommends that oral health systems orient their action towards disease prevention and primary health care, especially for disadvantaged and poor populations, rather than only towards disease treatment [2,11].

Dentistry in Italy is mostly private. Public dentistry exists but it is regulated by the Ministry of Health through the identification of so called “assistance essential levels” (LEA), that, concerning dentistry, offer selected basic dental care services to well-defined categories of citizens. Children are eligible for public dental care between 0 and 14 years of age, but only regarding selected basic treatments. As a consequence, Italy lacks a structured national oral health promotion program for children and delegates dental care, for both children and adults, mainly to private practices and the willingness of families [14].

In 2015, the Italian Statistics Agency (ISTAT) [14] reported that only 37.9% of the Italian population had visited a dentist in 2013 (the year of the investigation, performed on a sample of 60,000 citizens). The proportion reached 46.9% between 6 and 14 years of age. Furthermore, the same report stated that only 5% of dental care treatments were covered by public dentistry. Concerning oral hygiene habits, 73% of the sample reported to brush their teeth twice a day and 22.2% once a day, while 4.8% did so seldom or never. Recent, smaller scale Italian surveys have confirmed that children go to the dentist more often than adults. On the other hand, the knowledge on the causes of oral disease such as caries and periodontal inflammation and on how to prevent them among kids and their families is scarce [15]. As a result, it can be assumed that a large part of the Italian population doesn’t receive adequate information and support on maintaining their oral health. Furthermore, school-based oral health programs in Italy are not systematically included in the educative curriculum: some schools arrange their own programs with associations or professionals, but interventional actions, like screening, fluoride administration, mouth-rinses or varnishes cannot be performed at school.

The aims of this study were (i) to investigate, through a self-administered questionnaire, oral disease, oral health-related quality of life (OHRQoL), habits, socio-economic variables and knowledge concerning oral health in a sample of families whose children attended the public school summer services in Milan in 2018 and (ii) to assess the effects of lifestyle-related factors on oral diseases and OHRQoL.

## 2. Materials and Methods

This is a cross-sectional observational study, that served as pilot for a larger project designed by the Municipality of Milan (Italy) for the promotion of oral health in children.

### 2.1. Study Population 

In Italy education is compulsory from 6 to 16 years of age and schools close for the summer holidays between the beginning of June and the beginning of September. The city of Milan offers public summer camps for its students aged 6 to 11 during this period.

The city of Milan has about 1,400,000 inhabitants including about 20% immigrants [16] (compared to less than 10% in Italy) and is divided into nine municipalities. The characteristics of the population from different areas of the city can vary largely. The population’s declared income is about 33,000 euro/year on average (compared to about 23,000 euro in Italy) and presents a large variability, as about 35% of the residents declare less than 15,000 euro, about one fourth between 15,000 and 26,000, about one fourth between 26,000 and 55,000 euro and the remaining 12% over 55,000 euro per year. The sample population could not be preselected with respect to geographic origin or socio-economic status, due to strict privacy regulations concerning the use of sensitive data. For this reason, we extracted one summer school per municipality (nine in total), with a random process out of 41 total schools available, in order to study a sample population representative of the variability of the families of Milan.

### 2.2. Study Design 

In 2018, the Municipality of Milan organized a one-day educational program at the schools with the aim of providing oral health information to schoolchildren. A team of one dentist and one dental hygienist explained how to preserve good oral health and prevent oral diseases to the children, in every school, over the two weeks period. The intervention was designed as a game in which the children were invited to play the roles of bacteria, sugar, candies, soft drinks, juices, water, a toothbrush and fluoride. At the end of the lesson, the children were given stickers with simple rules on how to preserve oral health. A leaflet on oral health targeted to their parents was also provided.

No clinical examination of children was planned by the educational program. On the occasion of this one-day meeting, a questionnaire was given to the children’s caregivers, that included questions on different domains: oral diseases, oral health-related quality of life (OHRQoL), oral hygiene habits, food consumption, economic status, educational status. In the present study data from these questionnaires were analyzed.

### 2.3. Questionnaire Description and Administration

No open questions were included. A section of the questionnaire was aimed at evaluating the socio-emotional influence of oral health issues on the children and families and were adapted from the Child Oral Health Impact Profile (COHIP), for which reliability and validity have been tested extensively [17,18]. The questionnaire was delivered to the families of the children attending the schools in July 2018 by their teachers. The families had two weeks to answer the survey and handed it in anonymously at school. Summer school shifts last two weeks and one whole shift was considered. The teachers gave daily reminders on the importance of answering and returning the survey. The activity (survey) was explained to the parents at the meeting preceding the start of the summer camp. The families were informed on the goal of the study, told that the survey was meant to be filled in anonymously and that the data would be treated and analyzed by the Municipality of Milan and the University of Milan. The same information was included in an opening statement on the front page of the survey. By returning the survey, the consent was therefore granted.

### 2.4. Data Collection

In the present study, the first section of the questionnaire evaluated the children’s:Disease: current or past presence of tooth decay or tooth related abscess (yes/no);OHRQoL: current/past toothache or loss of school days due to toothache (yes/no).

Lifestyle-related variables of the children and their families were investigated in the following sections:Feeding habits: breastfeeding, the use of a sugared tea bottle or sugared pacifier (yes/no); snacks and juice/soft drinks intake (frequency scale).Oral hygiene protective behaviors: the habit to brush their teeth before going to school or to sleep and fluoride toothpaste use (frequency scale), toothbrush use initiation age and whether they received help from their family in brushing (scale).Dental coaching: professional support by pediatrician or dentist for the prevention of oral diseases (yes/no).Socio-economic and educational status: educational level, family income and the relevance of dental care cost on planning dental visits (scale). Summer school fees depend on family income measured on the ISEE (equivalent family income).

### 2.5. Statistical Analysis

We calculated odds ratios (OR) and 95% confidence intervals (CI) for the binary outcomes disease and OHRQoL using both univariate and multivariate (adjusting for sex and age from 6 to 11 years) unconditional logistic regression, chi-square tests and chi-square tests for trends in proportions with an alpha of 0.05 were carried on ordered categorical variables when effects were successively bigger or smaller. The R-project statistical software (R Foundation for Statistical Computing, Vienna, Austria) was used to carry out the analyses.

## 3. Results

### 3.1. Description of the Participants

The whole 41 school population in 2018 was 3819 children while the sample population of the 9 selected schools was 478 (12.5%). A total of 478 surveys were delivered and 296 of these (62%) were returned. Children of the analyzed population were aged 6 to 11 years. The survey compilers were 18% male and 82% women, and their age was: <30 years (6%), 31–45 years (67%), 46–60 years (26%) and >60 years (1%). The ethnic origin of the families was: 64% Italian, 10.5% Asian, 9.5% American, 9% European and 7% African. The children were 47% boys and 53% girls. Their age was 6 years (12%), 7 years (29%), 8 years (25%) 9 years (15%), 10 years (15%) and 11 years (4%).

### 3.2. Disease and OHRQoL

According to their parents a total of 118 children (40%) had experienced tooth decay or a tooth related abscess. In particular, 39% of the children had experienced at least one decayed tooth and 11% had had an abscess.

With regards to OHRQoL, a total of 82 children (28%) had experienced toothache or had lost school days due to toothache. In particular, among all children, 27% had suffered from toothache and 9% of the children had lost one or more days at school because of toothache or oral infections.

### 3.3. Lifestyle-Related Variables

Table 1, Table 2, Table 3 and Table 4 report data on the investigated lifestyle-related variables.

To assess how these lifestyle-related variables affect the oral disease and OHRQoL, with respect to the original questionnaire, the following variable categories were aggregated due to numerosity issues:juice/soft drinks consumption: never and sometimes avoiding were aggregated;toothbrushing before school, toothbrushing before going to bed and fluoride toothpaste use: “never” and “sometimes” were aggregated;first dental visit age aggregated to: 0–5, 6–7, and 8–11 years;dental care cost relevance: none and low were aggregated;educational level: none, elementary and middle school were aggregated;equivalent economic status indicator (ISEE): does not answer was considered missing.

#### 3.3.1. Effects of Variables on Disease

The effects of lifestyle-related variables on oral disease were calculated with multivariate unconditional logistic regression analysis and are reported in Figure 1.

Feeding habits: “Not consuming juice/soft drinks out of meals” showed a statistically significant protective OR both in the univariate model (0.47, CI 0.24–0.89) and in the adjusted model (0.52, CI 0.26–0.995) (*p* = 0.05) compared to “never” and “rarely”, while the “generally not consuming” category was not significant. However, the chi-squared test for trend in proportions was significant (*p* = 0.02), indicating a progressively more protective effect of avoiding out of meal juice. 

Protective hygiene behaviors: “Using fluoride toothpaste” also showed progressively more protective statistically significant ORs 0.39 and 0.36 (adjusted model) for “usually” and “always” vs “rarely” and “never” use fluoride toothpaste, trend (*p* = 0.009).

Dental coaching: No significant outcome was found for this variable.

Socio-economic and educational status: “Educational level” was statistically significant with successively stronger effects, however the proportional trend test was not significant (*p* = 0.08). The adjusted ORs for high school diploma and degree compared to none up to middle school were 0.52 (CI 0.28–0.97) and 0.47 (CI 0.24–0.91) respectively. The ISEE showed a significant proportional trend test (*p* = 0.007), however it only had a significant OR for the highest earning category (greater than 27,000 €) compared to the lowest earning one in the adjusted model (OR 0.43, CI 0.19–0.96).

#### 3.3.2. Effects of Variables on OHRQoL

The effects of lifestyle-related variables on OHRQoL were calculated with multivariate unconditional logistic regression analysis and are reported in Figure 2.

Feeding habits: The variables “tisane in a baby bottle” and “sweetened pacifier” were both statistically significant risk factors with adjusted ORs of 2.17 (CI 1.23–3.81) and 4.72 (CI 1.84–12.73) respectively. “Always avoiding juice/soft drinks” out of meals was protective (OR 0.33, CI 0.14–0.72) compared to “never avoiding”, while the “usually avoiding” category was not significant. However, the test for trend in proportions was significant (*p* = 0.002). Limiting junk food showed progressively greater protection (ORs 0.73, 0.65 and 0.48 adjusted, trend *p* = 0.019), however the ORs from the logistic models were not statistically significant even though very close, particularly in the univariate model.

Protective hygiene behaviors: The use of fluoride toothpaste was protective. Multivariate OR for “always use” compared to “never” or “rarely use” was 0.42 (CI 0.18–0.99), but the “usually use” fluoride toothpaste compared to “never” or “rarely use” was not significant. However, the trend test was significant (*p* = 0.002) indicating a progressive effect. Delaying the start of dental hygiene daily procedures compared to starting at tooth eruption, showed progressively greater ORs with a significant trend test (*p* = 0.026). However, only starting to brush teeth between 24 and 36 months compared to at tooth eruption was statistically significant and only in the univariate model (OR 2.17, CI 1.05–4.75). Helping the child with brushing their teeth also showed a progressive negative effect from always to never helping although the proportion trend test did not quite make it to statistical significance (*p* = 0.051). However, only the “never helping” compared to the “always helping” showed a significant OR, and only in the non-adjusted model (OR 2.36, IC 1.05–5.61).

Dental coaching: No significant association was found for this outcome variable.

Socio-economic and educational status: The importance of the cost of dental care and whether visits to the dentist would be more frequent if they costed less, were both progressively negatively associated with the OHRQoL outcome, with statistically significant trend tests (*p* = 0.029 and *p* = 0.006 respectively). Oral care cost importance was only significantly associated in the “very high” category compared to the low or no importance ones and only in the unadjusted estimate (OR 2.25, CI 1.11–4.7). Otherwise, the “more frequent oral visits if they cost less” was significantly associated in both the agree and strongly agree categories (ORs: 2.29, IC 1.03–5.42 and 2.70, IC 1.19–6.51 adjusted, respectively). Educational level showed a protective association in the adjusted model: ORs 0.49 (IC 0.25–0.95) for diploma and 0.47 (IC 0.23–0.95) for university degree versus none up to middle school. However, in spite of the successively greater protective effects the trend test was not statistically significant. The ISEE variable showed a strongly significant proportional chi-squared trend test (*p* = 0.003) with effects clearly stronger in succession. However, the OR was only significantly protectively associated in the most wealthy category (greater than 27,000 €) OR 0.28 (IC 0.09–0.70).

Numerosity, percentages chi-square test and chi-square proportion trend test results are given in Appendix A and the complete results for the logistic model analyses for both univariate and adjusted models are given in Appendix A.

## 4. Discussion

The present study reports the results of a survey on the oral and food habits of 296 families of children attending the public school summer services of the nine municipalities of the city of Milan. The sample population could not be preselected with respect to geographic origin or socio-economic status, due to strict privacy regulations concerning the use of sensitive data, therefore we selected one school from each municipality. On the other hand, the high response rate makes the results quite safe in terms of representativeness of the population’s differences, as 62% of the families completed and returned the questionnaire.

In the present study, 39% of the children up to 11 years of age, were reported to have experienced caries, 27% toothache, 11% an abscess and 8% had missed at least a school day due to dental issues. A study including 2603 4–5 years old pre-schoolers, out of 8323 residents in a northern Italian community, reported a caries prevalence and DMFT increasing from 17% and 0.5 at age 3 to 35% and 1.3 at age 5 and 85.8% of untreated carious lesions [19]. In another study on 5 year-old children, data collected always in northern Italy (Veneto), reported a DMFT of 1.3 and 68% of caries free [20]. In 12 year-olds, data published more than 10 years ago on Italian children, reported a DMFT ranging between 1.09 and 1.44 with a percentage of caries free children between 55.1 and 56.9% [21].

A study [22] performed on >25,000 6–7 year-old first-grade German schoolchildren, resulted in 60.9% caries free. In another German study including 496 5-year-old pre-school children and 608 8-year-old primary school children, the observed caries prevalence in the primary dentition was 26.2% and 48.8% respectively [23]. A survey and oral examination study performed on 1843 12- and 15-year old schoolchildren in Spain reported a DMFT/dmft of 0.89 and 1.38 and a caries prevalence 39.6% and 51.7%, respectively [24]. A study performed in Valencia, Spain, on 70 out of 1200 schools in the region, including 1373 children in the 6, 12 and 15 year-old age cohorts, resulted in a caries prevalence of 30% at 6 years of age, 37.7% at 12 years and 43.6% at 15 years [25]. A Swedish study from Jönköping on about 500 randomly sampled individuals, reported that 35% of the 3-year-olds were caries-free in 1973, compared to 79% 40 years later [26]. Adolescents aged 10 and 15 years exhibited the most pronounced reduction in DFS on the occlusal surfaces. By 2013, more than 90% of the proximal caries lesions in 15-year-olds were initial lesions. About 85% of 15-year-olds had a DFS of ≤5, whilst 1% exhibited a DFS of ≥26. The corresponding figures for 1973 were 0 and 45% respectively. The DFS score for the 20-year-olds was 35.1 in 1973 and 5.8 in 2013. Caries-free 20-year-olds were not seen until 1993 and reached 19% in 2013. The prevalence of caries in European children seems therefore to assess at over 30% (20 to 90% according to the European Regional Office of WHO [27]. A trend towards decreasing caries prevalence in school children has been described in several studies, nevertheless some authors have observed a plateau in the decline of caries prevalence in pre-schoolers in the recent years, especially in immigrant children or children from a low socio-economic level [19,27]. A limitation of our study consists in the fact that no oral examination was performed on the children and therefore the answers concerning disease prevalence could not be clinically confirmed. Nevertheless, despite being self-reported, caries prevalence is not so different from that detected when clusters of children were visited in other reports. Furthermore, recent studies have confirmed how questionnaires can be a valuable tool in gross assessment of the oral health status of children, as the answers provided by the caregivers concerning the occurrence of caries seem quite reliable, especially concerning advanced caries [28].

Caries are known to be associated to factors like *Streptococci mutans* transmission from the mother to the child, the consumption of frequent dietary sugars, poor tooth brushing, lack of topical fluoride applications and late dental examinations [29]. The results from the present study are in agreement with such evidence as a low consumption of juices, the use of a fluoride toothpaste and a higher educational and economical status were associated to protective effects towards oral diseases. Furthermore, the consumption of sweet tisanes in a baby bottle, the use of a sweetened pacifier and the concern regarding the cost of the dentist were associated to a negative effect of the oral status on life quality. Conversely, a low consumption of juices, a higher educational and economical status and seeing a dentist earlier in life had a positive impact on the OHRQoL.

Frequent sugar consumption is an established risk factor for caries development [30]. A study performed in England, Wales and Northern Ireland, investigating the association between consumption frequency of foods and drinks with added sugar and dental caries experience in the permanent teeth of 4950 12- and 15-year-old children [31], has shown how the consumption frequency of added sugars was associated with dental caries. A recent review [32] focused on an interesting aspect, that is the relationship between amount vs frequency of sugar restriction and caries prevention. The authors concluded that though amount and frequency are often strictly related, frequency reduction seems to be more effective and more easily achievable for patients in caries preventive programmes. Such simple and basic information on the risk and preventive factors associated with caries and oral diseases are not always understood by the population and the fact that only about half of the sample in the present study received information from health professionals on how to preserve their children’s oral health, is an issue. Actually, in the present study, 19% of the children had never been taken to a dentist, 43% of the children were seen by a dentist between the age of 3 and 5 years and 41% between 6 and 7 years. An early visit was associated to a positive effect of life quality related to oral health. In 2017, the Italian Academy of Conservative Dentistry (AIC) interviewed 1000 families with children < 14 years [15] to find out that more than 50% of the parents didn’t know that an excessive sugar intake or poor oral hygiene can affect caries onset and more than half the sample believed that decayed deciduous teeth don’t need to be treated. A recent Cochrane review [33] on the effect of interventions on pregnant women, new mothers and caregivers for the prevention of early childhood caries, has concluded that “Moderate-certainty evidence suggests that providing advice on diet and feeding to pregnant women, mothers or other caregivers with children up to the age of one year probably leads to a slightly reduced risk of early childhood caries”. In a German study investigating sweets consumption in preschoolers, it was observed that most children consumed sweets every day [34]. Further, sweets consumption was associated with cultural and contextual factors such as immigrant background, parental education, specific nutritional knowledge levels, and access arrangements in the home. The authors concluded that “dental practitioners should place more emphasis on gathering information from young patients regarding excessive and frequent consumption of sweets and consequently on trying to educate the children and their parents on oral health risks associated with such consumption. Particular attention is to be paid to children of Turkish and Arabic descent, as they have been shown to consume above-average amounts of sweets”. Similar results were observed in a Swedish study on caries prevalence and background factors in 4-year-old children: the authors observed that although the proportion of children with caries declined between 2007–2012, this decline applied only to non-immigrant children and prompted for a greater effort to change oral health behaviors, specifically for immigrant children.

A Cochrane review from 2016 investigating community-based population-level interventions for promoting child oral health, provided low evidence that community-based oral health promotion interventions that combine oral health education with supervised toothbrushing or professional preventive oral care can reduce dental caries in children, while interventions to promote access to fluoride, improve children’s diets or provide oral health education alone, had a limited impact [35]. The above reported results prompt well-studied and effective population-based interventions on pregnant women and caregivers in order to promote oral health in children and at the same time, the improvement of university programmes for general practitioners, pediatricians, dentists and dental hygienist regarding oral health advice and counseling.

In the present study the caregivers reported that their children brushed their teeth at least once a day in about 80% of the cases. Five percent did not use fluoridated products. Twenty-four% of the children had started brushing/cleaning from the time of eruption while 7% did so after 4 years of age. The children were always or sometimes assisted in brushing in 20% and 53% of the cases, respectively.

A controlled exposure to fluoride is a clear protective factor for caries prevention. The already mentioned review by van Loveren [30] reports how fluoride, appropriately used, reduces or breaks the relation between sugar consumption and caries. A Cochrane Review from 2019 has reported the benefits of using fluoride toothpaste in preventing caries when compared to non-fluoride toothpaste [36]. Hong et al. (2018) [31], have observed that, despite children who consume foods and drinks with added sugar more frequently are more likely to develop dental caries, a higher consumption frequency of drinking water in fluoridated areas might mitigate dental caries onset. Fluoride exposure through fluoridated drinking water has been reported to play a positive role in modifying the association between a long breastfeeding duration and caries. In a Korean study comparing the prevalence of caries in children living in a water-fluoridated and non-water-fluoridated community, the authors observed a lower dental caries prevalence in the fluoridated community. Even more interesting, when investigating the role of educational and socio-economic status on caries, the authors observed that differences in dental caries prevalence based on educational level were found in the non-fluoridated area but not in the fluoridated ones. The same concept applied for the socio-economic factors: fluoride community programs might help to reduce oral health inequalities among children [37].

Some studies in school children have shown how their oral hygiene routine is not always adequate. On the other hand, interventions like insisting on increasing the brushing time or providing children and their parents with individual tooth brushing instructions [38,39] have proven beneficial in oral hygiene parameters.

In the present study, a higher socio-economic status was confirmed to act as a protective factor towards oral diseases and related life quality. Non-regular dental check-ups have been identified, together with other behaviors, as a risk factor for caries in the permanent teeth of 4950 12- and 15-year-old children in England, Wales and Northern Ireland, using the Children’s Dental Health Survey 2013 (CDHS) data [30]. A study performed in Campania (Italy) on 5- and 12-year-old schoolchildren aiming at investigating the relationship between social and behavioral factors and caries experience, has shown a positive correlation with, among others, family income and mother’s educational level [40]. A similar correlation concerning the association between caries in deciduous teeth and mother’s geographical background was the result of a southern Italian study [27] performed on 513 preschool children 3 to 6 years of age in which the caries prevalence was estimated at 18.4%. In a study from Perugia (Italy) on 231 children aged 4–14 years [41], low familiar income and low parental education were both related to an increased prevalence of caries. A Dutch study including 630 5- to 6-year old children recruited from six large pediatric clinics in the country, investigating the relationship between family functioning and childhood dental caries, resulted in a lower DMFT and a better engagement in oral hygiene measures in children in functional and highly organized families. Disfunctional and less organized families often included lower educated mothers and immigrant background [42]. Another Dutch cross-sectional study including 5189 six-year-olds, investigated the role of parental education and employment status, household income, single parenting and teenage pregnancy in dental caries prevalence. The comparison among children without caries to children with mild or severe caries, resulted in an association between maternal education level and mild caries, while factors like low paternal educational level, parental employment status and household income played a role in children with severe caries. Furthermore, living in more disadvantaged districts was significantly associated with higher odds of dental caries.

The Iowa fluoride cohort study included a birth cohort from 1992 to 1995 and gathered dietary, fluoride and behavioral data at least twice yearly since recruitment plus oral examinations at ages 5, 9, 13 and 17 years. In 396 subjects with caries data available at ages 9, 13, and 17, Warren and co-workers found an association between high caries incidence with lower maternal educational level, less frequent tooth brushing, lower 100% juice consumption and being female [43]. An American study investigating parenting style and the demographics related to the oral health status of 132 children aged three to six years presenting to Nationwide Children’s Hospital dental clinic for an initial examination/hygiene appointment, observed that children with authoritative parents, a private dental insurance and attending daycare exhibited more positive behavior and/or less caries compared to children with authoritarian and permissive parents [44]. A review from 2015 [45] aimed to systematically assess evidence for the association between socioeconomic position and caries concluded that a low socioeconomic status “is associated with a higher risk of having caries lesions or experience” and found such association was even stronger in developed countries. A Swedish study from 2010 [46], investigating the influence of child and parental migration background on the risk of caries increment in 15538 Swedish adolescents, revealed that adolescents with foreign-born parents, irrespective of whether born in Sweden or abroad, showed a significantly elevated risk for approximal caries increment and developed more caries lesions compared to adolescents with Swedish-born parents. A recent Italian study [47] aiming at the evaluation of DMFT index, caries prevalence and Unmet Restorative Treatment Needs (UNT) index in migrant and not migrant children with low income on 553 12–14 years old children showed how economically disadvantaged children had experienced more caries and caries prevalence, and unmet restorative treatment needs index among migrant children was higher than that of non-migrants. In a study on Brazilian children aiming at investigating the relationship between early childhood caries and obesity in children from low-income families, the average body mass index of the children was not associated with dental caries. On the other hand, the lack of caries experience in children was associated with a higher family income [48].

The present study included questions concerning OHRQoL. OHRQoL is a multidimensional construct that includes a subjective evaluation of the individual’s oral health, functional well-being, emotional well-being, expectations and satisfaction with care, and sense of self [49]. In the present study, the questionnaire was prepared to investigate several aspects (biological, environmental and relating the everyday life) that are directly or indirectly related to oral health and school environment. The investigated field is wide and to avoid preparing a too large questionnaire that would have limited the compliance of caregivers for its compilation, limited number of the most significant questions was selected for each field. Regarding the OHRQoL the questions that were more adherent to the purpose of the project were chosen.

From the answers, it resulted that 82 children (28%) had experienced toothache or had lost school days due to toothache. As mentioned in the introduction, children with a poor oral health or limited access to dental care facilities, might have an impaired activity and even have an impact on lost schooldays, as confirmed by the present study [8,9]. A systematic review on the influence of parental socio-economic status and home environment on children’s OHRQoL, reported that the majority of the studies resulted in a better OHRQoL in children from families with high income, parental education and family economy and indicated mothers’ age, family structure, household crowding and presence of siblings as significant predictors of children’s OHRQoL [50].

Socio-economic disparities concerning OHRQoL have been observed in Canadian children from low-income families compared to higher income families [51] as well as sociodemographic factors have proven to be strongly associated with low OHRQoL in the US, where oral health disparities and reduced OHRQoL have been found to be more prevalent among racial and ethnic minorities and those with socio-economic disadvantages. In a Brazilian study on 1632 5-year-old preschool children [52], the impact of oral health conditions on oral health-related quality of life was evaluated by means of a survey. Dental caries experience had a negative impact on OHRQoL and in particular, families with low income and younger parents reported a greater impact. Furthermore, according to parents’ perceptions, a poor general health status rating was related to poorer quality of life among the children. Similar results have been confirmed by other Brazilian studies [53,54,55]. In particular, Chaffee and co-workers observed that, being the level of caries experience similar, disadvantaged families reported a larger impact of oral diseases on their life quality compared to wealthier families, suggesting that quality of life perception may be subjective and variable in different socio-economic groups. A population-based case-control study including 546 Brazilian children aged 8–10 years (182 cases with a high negative impact on OHRQoL and 364 controls with a low negative impact on OHRQoL), investigating the effect of dental caries experience, malocclusion, and traumatic dental injuries on OHRQoL, reported no significant difference in traumatic dental injuries and malocclusion between the case and control groups, while children with DMFT/dmft ≥ 3 had a 2.06-fold greater chance of experiencing a high negative impact on OHRQoL than those with DMFT/dmft = 0 [56].

Many studies confirm a negative impact of dental caries on OHRQoL. In a recent study from Hong Kong on 336 preschool children, assessing the relation between oral status and OHRQoL, caries experience was reported to be associated with lower OHRQoL, while parents’ education level was not [57]. A study involving 5484 North Carolina schoolchildren and their families, carried out to determine the impact of enamel fluorosis, beyond dental caries, on OHRQoL, reported no association between fluorosis and OHRQoL, while a child’s caries experience negatively affected OHRQoL [58].

A Dutch population-based prospective cohort study investigated, in 2833 children, the relation between dental caries at the age of 6 with OHRQOL assessed at the age of 10. The authors reported that “the higher the dmft-score at the age of 6, the lower the OHRQOL at the age of 10 (*p* < 0.001)” and stressed the importance of oral health care during childhood to prevent a poor OHRQOL later in life, as OHRQOL seems not only to be related to current oral health status, but also to oral health experiences from the past [59].

An interesting finding in this respect concerning OHRQoL was reported by Merdad and co-workers who, in a cross-sectional study on 1312 middle school children from Jeddah, Saudi Arabia, observed how dental pain as the reason for the most recent dental visit was associated with poor OHRQoL, while receiving a filling during the previous dental visits was significantly associated with better OHRQoL. Other factors related to a poor OHRQoL in the Saudi Arabian context were: having a larger number of siblings, a lower family income, a lower paternal education level, health problems and prior hospitalization [60].

As a matter of fact, OHRQoL is included in the World Health Organization’s Global Oral Health Program (2003). Though it has been shown that children and caregiver ratings on OHRQoL can be in disagreement, with caregivers rating their children’s QoL lower than the children themselves, in the present study the children were not interviewed concerning their own perception as regulations did not allow it [61].

## 5. Conclusions

In conclusion, protective behaviors and socio-economic status affect oral diseases and OHRQoL in children of Milan. Such results suggest that families and caregivers could profit from a deeper knowledge on the most common oral diseases concerning protective behaviors that could help preventing their onset. Health professionals like gynecologists, pediatricians, dentists, dental hygienist and general practitioners may collaborate in conveying such information consistently and effectively. Furthermore, data of the present study is the ground on which future prevention programmes of oral health would be designed for pre-school and school children in Milan. A main limitation of this study is that data on caries history in our sample was based on questionnaires only and the lack of a clinical examination might imply a reduced accuracy in caries detection. An effort will be done to overcome such issue in the future.

## Figures and Tables

**Figure 1 ijerph-17-06612-f001:**
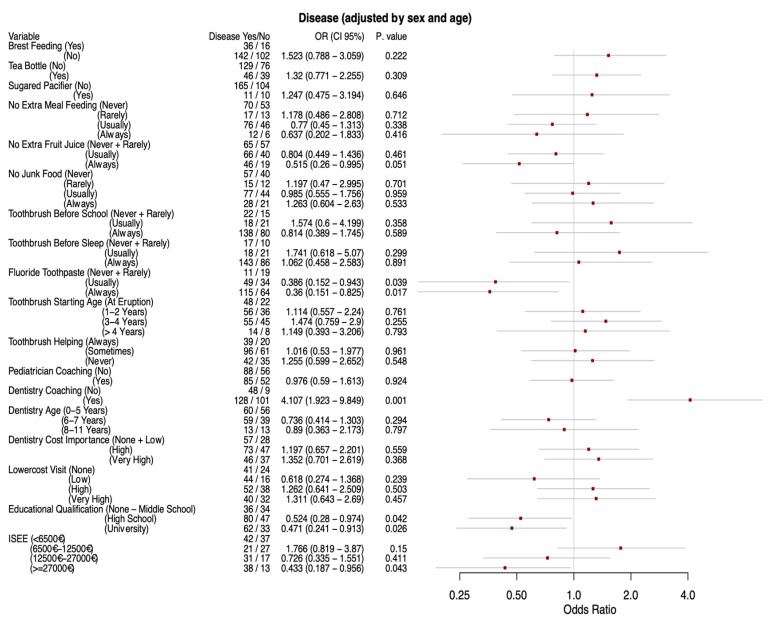
Multivariate unconditional logistic regression analysis for the outcome disease adjusted by sex and age.

**Figure 2 ijerph-17-06612-f002:**
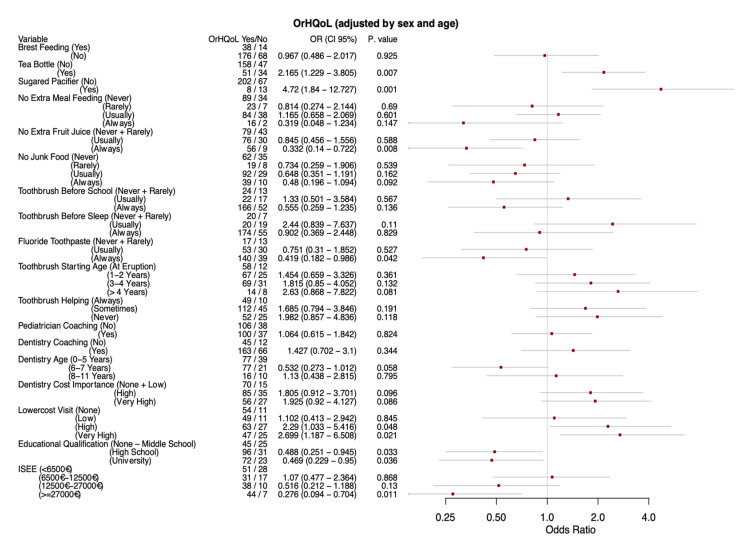
Multivariate unconditional logistic regression analysis for the outcome OHRQoL.

**Table 1 ijerph-17-06612-t001:** Questions and data on feeding habits.

Feeding Habits					
	**no/never**	**rarely**	**usually**	**yes/always**	***n*** **/a**
**Has your child been breastfed?**	18% (*n* = 53)	/	/	82% (*n* = 243)	/
**Have you ever dipped a pacifier in anything sweet to calm down your child?**	93% (*n* = 275)	/	/	7% (*n* = 21)	/
**Did you use to administer sweet tisanes to your child through a baby bottle?**	71% (*n* = 210)	/	/	29% (*n* = 86)	/
**Does your child avoid to eat between meals (snack)?**	10% (*n* = 30)	42% (*n* = 123)	41% (*n* = 122)	6% (*n* = 18)	1% (*n* = 3)
**Does your child avoid to drink juices/soft drinks between meals?**	7% (*n* = 21)	34% (*n* = 101)	36% (*n* = 106)	22% (*n* = 65)	1% (*n* = 3)
**Does your child avoid to eat candies and chocolate (junk food) frequently?**	9% (*n* = 27)	33% (*n* = 97)	41% (*n* = 122)	17 (*n* = 50)	0

**Table 2 ijerph-17-06612-t002:** Questions and data on protective hygiene behaviors.

Protective Hygiene Behaviors					
	**no/never**	**rarely**	**usually**	**yes/always**	***n*** **/a**
**Does your child brush her/his teeth every morning before school?**	4% (*n* = 11)	9% (*n* = 26)	13% (*n* = 39)	74% (*n* = 219)	0
**Does your child brush her/his teeth every evening before bed?**	0	9% (*n* = 27)	13% (*n* = 39)	78% (*n* = 229)	0
**Does your child brush her/his teeth with a fluoride toothpaste?**	5% (*n* = 16)	5% (*n* = 14)	28% (*n* = 83)	60% (*n* = 179)	1% (*n* = 4)
	**at eruption**	**at 1–2 years**	**at 2–3 years**	**>4 years**	***n*** **/a**
**When did your child start to brush her/his teeth?**	24% (*n* = 70)	31% (*n* = 92)	34% (*n* = 100)	7% (*n* = 22)	4% (*n* = 12)
	**no/never**	**sometimes**		**yes/always**	***n*** **/a**
**Do you help your child to brush her/his teeth?**	26% (*n* = 77)	53% (*n* = 157)		20% (*n* = 59)	1% (*n* = 3)

**Table 3 ijerph-17-06612-t003:** Questions and data on dental coaching.

Dental Coaching					
	**no/never**		**yes/always**		
**Has your pediatrician even advised you on oral prevention or hygiene?**	51% (*n* = 151)		49% (*n* = 145)		
**Has your child already gone to the dentist?**	19% (*n* = 57)		81% (*n* = 239)		
	**<3 years**	**3–5 years**	**6–7 years**	**8–9 years**	**10–11 years**
**If your child has already gone to the dentist, how old was her/his at the first dental visit?**	6% (*n* = 14)	42% (*n* = 101)	41% (*n* = 98)	9% (*n* = 21)	2% (*n* = 5)

**Table 4 ijerph-17-06612-t004:** Questions and data on socio-economic and educational qualification.

Socio-Economic and Educational Qualification					
	**not/none**	**low**	**high**	**very high**	***n*** **/a**
**In planning any dental visits for your child, how important their cost can be?**	8% (*n* = 23)	21% (*n* = 62)	40% (*n* = 120)	28% (*n* = 83)	3% (*n* = 8)
**If the cost of the dentist was lower, I would take my daughter to visit more often. Do you agree?**	22% (*n* = 65)	20.3% (*n* = 60)	30.4% (*n* = 90)	24.3% (*n* = 72)	3% (*n* = 9)
	**none**	**elementary school**	**middle school**	**high school diploma**	**graduated from university**
**What is the educational qualification of the compiler?**	3% (*n* = 9)	3% (*n* = 9)	18% (*n* = 52)	43% (*n* = 126)	33% (*n* = 96)
	**<6500 euros**	**6500 to 12,500 euros**	**12,500 to 27,000 euros**	**>27,000 euros**	***n*** **/a**
**ISEE**	28% (*n* = 84)	17% (*n* = 50)	17% (*n* = 51)	19% (*n* = 55)	19% (*n* = 56)

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
