# Peer review of "Impact of Lifestyle Variables on Oral Diseases and Oral Health-Related Quality of Life in Children of Milan (Italy)"

_ijerph, 2020, doi:10.3390/ijerph17186612_

Round 1
Reviewer 1 Report
page 1-list of authors-correct font style and size
page 3 line 92, "however" should be erased
page 3-OHRQoL is first time mentioned in main text and should be explained.
page 3 line 100-space after "apilot"- a pilot
page 3 line 114 less space in (9 total)
page 4 line 153 "t" should be erased
page 9, line 296-302, there is no reference number for qoute Jönköping..Not in the References also
page 9, line 276 , change to 62%
Conclusions- add limitations of this study;only questionarries were performed without clinical analysis-lack of accurate caries indexes
References list should be checked and corrected according to Journal instructions
I would a
page 11 line 368, change to 28%
Author Response
Dear reviewer,
thank you for your kind revision of the manuscript. We modified the entire manuscript following your indications, but we did not perform the "page 9, line 276 , change to 62%" and "page 11 line 368, change to 28%" since we cannot find the line. Could you please indicate the sentence?
The answer to each issue is reported below.
We think that the manuscript is largely improved.
kind regards
Daniela Carmagnola,
Gaia Pellegrini,
Matteo Malvezzi,
Elena Canciani,
Dolaji Henin
Claudia Dellavia
Comments and Suggestions for Authors
page 1-list of authors-correct font style and size
ANSWER: the entire document has been formatted.
page 3 line 92, "however" should be erased
ANSWER: “however” has been erased.
page 3-OHRQoL is first time mentioned in main text and should be explained.
ANSWER: the acronym has been explained
page 3 line 100-space after "apilot"- a pilot
ANSWER: the correction has been done.
page 3 line 114 less space in (9 total)
ANSWER: the correction has been done.
page 4 line 153 "t" should be erased
ANSWER: the correction has been done.
page 9, line 296-302, there is no reference number for qoute Jönköping..Not in the References also
ANSWER: the following reference has been added: koch, G.; Helkimo, A.N.; Ullbro, C. Caries Prevalence and Distribution in Individuals Aged 3-20 Years in Jönköping, Sweden: Trends Over 40 Years. Eur Arch Paediatr Dent 2017,18(5), 363-370.
page 9, line 276 , change to 62%
ANSWER: sorry, we cannot find the line. Could you please indicate the sentence?
Conclusions- add limitations of this study; only questionarries were performed without clinical analysis-lack of accurate caries indexes
ANSWER: conclusions were changed as follows: “Furthermore, data of the present study is the ground on which the following prevention programmes of oral health would be designed for pre-scholar and scholar children in Milan. A main limitation of this study is that data on caries history in our sample is based on questionnaires only and the lack of a clinical examination might imply a reduced accuracy in caries detection. An effort will be done to overcome such issue in the future.”
Reviewer 2 Report
Line 28-30: Concerning disease, the never consumption of fruit juice, the use of … IT MAY BE BETTER TO SAY ‘ WITH RESPECT TO DISEASE, THE ‘NEVER’ CONSUMPTION…
29 fluoride toothpaste, higher educational qualification and ISEE (equivalent family income) of those 30 who filled out the form resulted protective….. PERHAPS CHANGE TO… RESULTED IN PROTECTIVE
Line 30-34: Regarding OHRQoL, the never assumption/use of tea 31 bottle, sugared pacifier and fruit juice as well as the use of fluoride toothpaste, the higher ..PLACE NEVER IN INVERTED COMMAS ‘’NEVER’
32 educational qualification and ISEE of those who filled out the form resulted protective. INCORRECT AS IT STANDS NOW. PERHAPS CHANGE TO …PROTECTIVE EFFECTS?
LINE 33: Concluding, IT MAY BE BETTER TO SAY.. IN CONCLUSION…
33 protective behaviors and socio-economic status affect the oral disease and OHRQoL in children of Milan
Line 100: apilot….A PILOT
Page 197: never and sometimes… PERHAPS USE INVERTED COMMAS ‘NEVER’
Page 307: A limit of this study… A LIMITATION OF THE STUDY….
327-329: A recent review [24] pointing on one 31] 328 pointed out an interesting aspect, that is the relationship between amount vs frequency of sugar 329 restriction and caries prevention, hasit concluded that though am… THIS IS WRITTEN INCORRECTLY. MAYBE WRITE THIS IN TWO SENTENCES
353: and Arabic decent, DESCENT
Line 368: 5% TRY NOT TO START THE SENTENCE WITH A NUMBER. …Five pecent….
line 392: Non regular NON-REGULAR
LINE 406: well –organized HIGHLY ORGANISED MAY BE BETTER. ‘WELL’ MAY DENOTE A VALUE JUDGEMENT.
LINE 418 coworkers found…CO-WORKERS
LINE 494-495: In conclusion,Concluding, protective behaviors and socio-economic status affect the oral disease 495 and OHRQoL….. IN CONCLUSION
Author Response
Dear reviewer,
thank you for your kind revision of the manuscript. We modified the entire manuscript following your indications and the answer to each issue is reported below.
We think that the manuscript is largely improved.
kind regards
Daniela Carmagnola,
Gaia Pellegrini,
Matteo Malvezzi,
Elena Canciani,
Dolaji Henin
Claudia Dellavia
Comments and Suggestions for Authors
Line 28-30: Concerning disease, the never consumption of fruit juice, the use of … IT MAY BE BETTER TO SAY ‘ WITH RESPECT TO DISEASE, THE ‘NEVER’ CONSUMPTION…
ANSWER: the correction has been done.
29 fluoride toothpaste, higher educational qualification and ISEE (equivalent family income) of those 30 who filled out the form resulted protective….. PERHAPS CHANGE TO… RESULTED IN PROTECTIVE
ANSWER: the sentence has been modified as follows: “With respect to disease, the never consumption of fruit juice, the use of fluoride toothpaste, higher educational qualification and ISEE (equivalent family income) of those who filled out the form resulted protective factors.”
Line 30-34: Regarding OHRQoL, the never assumption/use of tea 31 bottle, sugared pacifier and fruit juice as well as the use of fluoride toothpaste, the higher ..PLACE NEVER IN INVERTED COMMAS ‘’NEVER’
ANSWER: the correction has been done.
32 educational qualification and ISEE of those who filled out the form resulted protective. INCORRECT AS IT STANDS NOW. PERHAPS CHANGE TO …PROTECTIVE EFFECTS?
ANSWER: the sentence has been modified as follows “the form resulted to have protective effects”
LINE 33: Concluding, IT MAY BE BETTER TO SAY.. IN CONCLUSION… protective behaviors and socio-economic status affect the oral disease and OHRQoL in children of Milan
ANSWER: the correction has been done.
Line 100: apilot….A PILOT
ANSWER: the correction has been done.
Page 197: never and sometimes… PERHAPS USE INVERTED COMMAS ‘NEVER’
ANSWER: the correction has been done.
Page 307: A limit of this study… A LIMITATION OF THE STUDY….
ANSWER: the correction has been done.
327-329: A recent review [24] pointing on one 31] 328 pointed out an interesting aspect, that is the relationship between amount vs frequency of sugar 329 restriction and caries prevention, hasit concluded that though am… THIS IS WRITTEN INCORRECTLY. MAYBE WRITE THIS IN TWO SENTENCES
ANSWER: the sentence has been modified has follows:
Frequent sugar consumption is an established risk factor for caries development [30]. A study performed in England, Wales and Northern Ireland, investigating the association between consumption frequency of foods and drinks with added sugar and dental caries experience in the permanent teeth of 4950 12- and 15-year-old children [31], has shown how the consumption frequency of added sugars was associated with dental caries. A recent review [32] focused on an interesting aspect, that is the relationship between amount vs frequency of sugar restriction and caries prevention. The authors concluded that though amount and frequency are often strictly related, frequency reduction seems to be more effective and more easily achievable for patients in caries preventive programmes.
353: and Arabic decent, DESCENT
ANSWER: the correction has been done.
Line 368: 5% TRY NOT TO START THE SENTENCE WITH A NUMBER. …Five pecent….
ANSWER: the correction has been done.
line 392: Non regular NON-REGULAR
ANSWER: the correction has been done.
LINE 406: well –organized HIGHLY ORGANISED MAY BE BETTER. ‘WELL’ MAY DENOTE A VALUE JUDGEMENT.
ANSWER: the correction has been done.
LINE 418 coworkers found…CO-WORKERS
ANSWER: the correction has been done.
LINE 494-495: In conclusion,Concluding, protective behaviors and socio-economic status affect the oral disease 495 and OHRQoL….. IN CONCLUSION
ANSWER: the correction has been done.
Reviewer 3 Report
Revision of the article:
Impact of lifestyle variables on oral diseases and oral health-related quality of life in children of Milan (Italy)
The following article in my opinion continues to have aspects to improve prior to its publication, such as:
- As I mentioned in the previous review of the article, the concept of oral health related quality of life (OHRQoL) is not adequately addressed. To assess the OHRQoL of children with oral diseases it is not enough to measure the current/past toothache or loss of school days due to toothache with a Yes o No rating. To assess how children and parents perceive the impact of oral diseases on their quality of life or different aspects of life, another type of assessment and measurement is needed. It is not enough to assess the two aspects measured in this study: the presence of toothache and the loss of school days. The study does not explain how oral diseases affect different parts of children´s life on a physical, psychological and social level. These alternative measures are in the form of standardized questionnaires, that have not been used in this study. You can see a definition of OHRQoL in the article: Sischo L, Broder HL. Oral Health-related Quality of Life. What, Why, How, and Future Implications. J Dent Res. 2011; 90(11): 1264-1270: “OHRQoL is a multidimensional construct that includes a subjective evaluation of the individual’s oral health, functional well-being, emotional well-being, expectations and satisfaction with care, and sense of self”. This is a definition based on the previous WHO definition of Health and Quality of Life.
- The abstract of the article follows without mentioning the study design.
- In the Keywords it is repeated twice: oral disease.
- In the section Materials and Methods, when the study design is mentioned, the term “cross-sectional” should be included to make it more complete.
- In the section Materials and Methods, in 2.2. Study design, it is not necessary to explain the one-day educational program in the sense that the study is carried out separately by providing the questionnaire to the parents.
- It is not explained why an own questionnaire was created, instead of using the questionnaires already validated and used in previous studies.
- Sections 2.3 and 2.4 could be joined.
- In lines 171-172 the information is repeated.
- In lines 184-186 the information is not clear in relation to the percentage of toothache: “With regards to OHRQoL, 82 children (28%) had experienced toothache or had lost school days 184 due to toothache. In particular, 27% had suffered from toothache and 9% of the children had lost one 185 or more days at school because of toothache or oral infections”.
- It is necessary to correct some errors in the References section. You can check the recommendations in
https://www.mdpi.com/journal/ijerph/instructions
https://www.mdpi.com/authors/references
I hope that the suggested changes help to improve the quality of the article and that they are well received.
Kind regards
Author Response
Dear reviewer,
thank you for your kind revision of the manuscript. We modified the entire manuscript following your indications. As reported below, we did not have the possibility to investigate all the multidimensional aspects that compose the OHRQoL. However, thanks to your comments, we had the opportunity to clarify and debate this important issue.
We think that the manuscript is largely improved following your comments.
kind regards
Daniela Carmagnola,
Gaia Pellegrini,
Matteo Malvezzi,
Elena Canciani,
Dolaji Henin
Claudia Dellavia
Revision of the article:
Impact of lifestyle variables on oral diseases and oral health-related quality of life in children of Milan (Italy)
The following article in my opinion continues to have aspects to improve prior to its publication, such as:
- As I mentioned in the previous review of the article, the concept of oral health related quality of life (OHRQoL) is not adequately addressed. To assess the OHRQoL of children with oral diseases it is not enough to measure the current/past toothache or loss of school days due to toothache with a Yes o No rating. To assess how children and parents perceive the impact of oral diseases on their quality of life or different aspects of life, another type of assessment and measurement is needed. It is not enough to assess the two aspects measured in this study: the presence of toothache and the loss of school days. The study does not explain how oral diseases affect different parts of children´s life on a physical, psychological and social level. These alternative measures are in the form of standardized questionnaires, that have not been used in this study. You can see a definition of OHRQoL in the article: Sischo L, Broder HL.Oral Health-related Quality of Life. What, Why, How, and Future Implications. J Dent Res. 2011; 90(11): 1264-1270: “OHRQoL is a multidimensional construct that includes a subjective evaluation of the individual’s oral health, functional well-being, emotional well-being, expectations and satisfaction with care, and sense of self”. This is a definition based on the previous WHO definition of Health and Quality of Life.
- ANSWER: thank you for arising this point that allows us to clarify and debate this important issue. The questionnaire of the present study has been administrated to children’s caregivers during the one-day educational program at the schools. Questions were prepared in order to investigate the several factors (biological, environmental and relating the everyday life) that are directly or indirectly related to oral health and school environment of children. The investigated field is really wide and to avoid preparing a too large survey that would have limited the compliance of caregivers for its compilation, we had to select a limited number of the most significant questions for each field. Despite this limitation, the survey allowed us to characterize this specific population thus planning the following interventions of promotion of oral health in children. Regarding the OHRQoL we choose the questions from COHIP that were more adherent to the purpose of the project. We know that the OHRQoL is a more complex and multidimensional construct and we would investigate it entirely in future more specific projects. However, the purpose and setting of this study did not allow us to completely investigate it. To discuss this issue, the following sentence has been added: "OHRQoL is a multidimensional construct that includes a subjective evaluation of the individual’s oral health, functional well-being, emotional well-being, expectations and satisfaction with care, and sense of self [49]. In the present study, the questionnaire was prepared to investigate several aspects (biological, environmental and relating the everyday life) that are directly or indirectly related to oral health and school environment. The investigated field is wide and to avoid preparing a too large questionnaire that would have limited the compliance of caregivers for its compilation, limited number of the most significant questions was selected for each field. Regarding the OHRQoL the questions from COHIP that were more adherent to the purpose of the project were chosen."
- The abstract of the article follows without mentioning the study design.
ANSWER: Abstract has been modified as follows: "A large part of the Italian population doesn’t receive adequate information and support on how to maintain oral health. In this observational, cross-sectional, pilot study, we investigated how some lifestyle-related variables affect oral disease and oral health-related quality of life (OHRQoL) of children attending public-school summer services in Milan”
- In the Keywords it is repeated twice: oral disease.
- ANSWER: the correction has been done.
- In the section Materials and Methods, when the study design is mentioned, the term “cross-sectional” should be included to make it more complete.
- ANSWER: the correction has been done.
- In the section Materials and Methods, in 2.2. Study design, it is not necessary to explain the one-day educational program in the sense that the study is carried out separately by providing the questionnaire to the parents.
- ANWER: following the indications of another reviewer, this part has been added to define the setting of the study.
- It is not explained why an own questionnaire was created, instead of using the questionnaires already validated and used in previous studies. ANSWER: this point has been discussed above.
- Sections 2.3 and 2.4 could be joined.
- ANSWER: sections 2.3 and 2.4 were organized following the suggestions of another reviewer.
- In lines 171-172 the information is repeated.
- ANSWER: sorry, we cannot find the line. Could you please indicate the sentence?
- In lines 184-186 the information is not clear in relation to the percentage of toothache: “With regards to OHRQoL, 82 children (28%) had experienced toothache or had lost school days 184 due to toothache. In particular, 27% had suffered from toothache and 9% of the children had lost one 185 or more days at school because of toothache or oral infections”.
ANSWER: the sentence has been modified as follows: “With regards to OHRQoL, a total of 82 children (28%) had experienced toothache or had lost school days due to toothache. In particular, among all children, the 27% had suffered from toothache and the 9% of the children had lost one or more days at school because of toothache or oral infections.
“
- It is necessary to correct some errors in the References section. You can check the recommendations in
- ANSWER: the reference section has been checked and modified.
I hope that the suggested changes help to improve the quality of the article and that they are well received.
Reviewer 4 Report
Thank you for giving me the opportunity to re-examine thesis in good material. The review of each item of thesis content seems to have been corrected. You did a lot of work.
Author Response
Dear reviewer,
thank you for revisited our manuscript and for your kind comment.
kind regards
Daniela Carmagnola,
Gaia Pellegrini,
Matteo Malvezzi,
Elena Canciani,
Dolaji Henin
Claudia Dellavia
This manuscript is a resubmission of an earlier submission. The following is a list of the peer review reports and author responses from that submission.
Round 1
Reviewer 1 Report
Dear Authors,
The manuscript is well written, well organized and technically prepared according to IJERPH instructions.
But, I have some concerns:
- The authors analyzed only questionaries’ from parents and did not analyzed the oral health status of children (DMFT, dmft..etc.). With this correlation, the study will give correct answers about caries prevalence in children and possible connection with oral health habits and socioeconomic status.
Nice examples of correlation between oral health status and environmental conditions, published in IJERPH:
Early Childhood Caries and Body Mass Index in Young Children from Low Income Families. Int. J. Environ. Res. Public Health 2013, 10(3), 867-878; https://doi.org/10.3390/ijerph10030867
Associations of Community Water Fluoridation with Caries Prevalence and Oral Health Inequality in Children. Int. J. Environ. Res. Public Health 2017, 14(6), 631; https://doi.org/10.3390/ijerph14060631
- Because this is pure observational study based on questionnaires, there are many bias and limits. First, the parents give "correct " answers into questionaries’ regarding diet, sugar consumption and tooth brushing. So, the data collected can be only auxiliary methods, helping in describing the clinical examination of selected children (caries indexes)
- The authors in discussion should quote more published paper about caries indexes in children in EU and Italy’s neighboring counties, resulting in more complex data about caries prevalence in children
Author Response
Dear reviewer, thank you for your review of the manuscript titled "
"Impact of lifestyle variables on oral diseases and oral health related quality of life: a survey in children aged 6 to 11 years attending summer schools at Milan, Italy".
Following your indications we made changes to the overall text. Here below, we report the detailed answers to the your suggestions. Main changes in the text are highlighted in green. The manuscript with revisions has been attached to this notes.
We think that the quality of the manuscript is widely improved.
Thank you for your consideration of our work for publication on IJERPH.
The manuscript is well written, well organized and technically prepared according to IJERPH instructions.
But, I have some concerns:
- The authors analyzed only questionaries’ from parents and did not analyzed the oral health status of children (DMFT, dmft..etc.). With this correlation, the study will give correct answers about caries prevalence in children and possible connection with oral health habits and socioeconomic status.
ANSWER: thank you for raising this issue. This observational study represents the pilot part of a larger project designed by the Municipality of Milan (Italy) for the promotion of oral health in children. In 2018, the Municipality of Milan organized a one day educational program at the schools with the aim of providing oral health information to the schoolchildren of Milan. A team of one dentist and one dental hygienist explained how to preserve good oral health and prevent oral diseases to the children, in every individual school over the 2 weeks period. On the occasion of this one-day meeting, the questionnaire was created for the children’s caregivers. No clinical examination of children wasplanned by the educational program and it was not possible to perform it. The study design has been clarified in methods and this limit has been discussed.
- Nice examples of correlation between oral health status and environmental conditions, published in IJERPH:
Early Childhood Caries and Body Mass Index in Young Children from Low Income Families. Int. J. Environ. Res. Public Health 2013, 10(3), 867-878; https://doi.org/10.3390/ijerph10030867
Associations of Community Water Fluoridation with Caries Prevalence and Oral Health Inequality in Children. Int. J. Environ. Res. Public Health 2017, 14(6), 631; https://doi.org/10.3390/ijerph14060631
ANSWER: these references have been added in the discussion
- Because this is pure observational study based on questionnaires, there are many bias and limits. First, the parents give "correct " answers into questionaries’ regarding diet, sugar consumption and tooth brushing. So, the data collected can be only auxiliary methods, helping in describing the clinical examination of selected children (caries indexes).
ANSWER: thank you. This issue has been discussed in the first point. This limit has been discussed in the discussion.
- The authors in discussion should quote more published paper about caries indexes in children in EU and Italy’s neighboring counties, resulting in more complex data about caries prevalence in children
Answer: several papers about caries indexes in children in EU and Italy’s neighboring counties have been added in discussion

Reviewer 2 Report
The language utilised in the paper needs
The topic is an important one. However the conceptualisation and design of the paper needs considerable improvement.
The introduction lacks and good theoretical grounding of oral health quality of life, children's oral health and behavioral interventions.
The methods are problematic.
The rationale for the study, the selection of this particular sample and its limitations are superficially dealt with. The sampling itself needs to be clarified, justified and its implications in terms of extrapolations need to be considered.
The validity and reliability of the instruments used are not considered leaving the methods questionable.
Ethical considerations, informed consent of children and guardians are not mentioned.
I would be hesitant, despite what seems like a rigorous analysis and application of statistical methods to assess the results in detail. also the discussion is also dependent on the framing of the introduction which would require considerable revision if the manuscript can be salvaged.
Does the paper anything new or novel which we already do not know? This remains unexplained in the paper
Author Response
Dear reviewer, thank you for your review of the manuscript titled "
"Impact of lifestyle variables on oral diseases and oral health related quality of life: a survey in children aged 6 to 11 years attending summer schools at Milan, Italy".
Following your indications we made changes to the overall text. Here below, we report the detailed answers to the your suggestions. Main changes in the text are highlighted in green. The revisited manucript has been attached to this notes.
We think that the quality of the manuscript is widely improved.
Thank you for your consideration of our work for publication on IJERPH.
REVIEWER 2
The language utilised in the paper needs.
ANSWER: the language has been checked by a native English speaker and technical writer
The topic is an important one. However the conceptualisation and design of the paper needs considerable improvement.
ANSWER: the study design, methods and results of the study have been rewritten to make them more understandable. Furthermore, limits of the study have been reported in the discussion.
The introduction lacks and good theoretical grounding of oral health quality of life, children's oral health and behavioral interventions.
ANSWER: the introduction has been rewritten to improve the theoretical grounding of oral health quality of life, children's oral health and behavioral interventions.
The methods are problematic. The rationale for the study, the selection of this particular sample and its limitations are superficially dealt with. The sampling itself needs to be clarified, justified and its implications in terms of extrapolations need to be considered.
ANSWER: the study population of summer school has been chosen to be as representative as possible of the families of Milan. To clarify this aspect, the following paragraph was added to the methods section:
“In Italy education is compulsory from 6 to 16 years of age and schools close for the summer holidays between the beginning of June and the beginning of September. The city of Milan offers public summer camps for its students aged 6 to 11 during this period.
The city of Milan has about 1.400.000 inhabitants including about 20% immigrants [8] (compared to less than 10% in Italy) and is divided in 9 municipalities. The characteristics of the population from different areas of the city, can vary largely. The population’s declared income is about 33.000 euro/year on average (compared to about 23.000 euro in Italy) and presents a large variability, as about 35% of the residents declare less than 15.000 euro, about one fourth between 15.000 and 26.000, about one fourth between 26.000 and 55.000 euro and the remaining 12% over 55.000 euro per year. The sample population could not be preselected with respect to geographic origin or socio-economic status, due to strict privacy regulations concerning the use of sensitive data, for this reason we randomly chose one summer school, out of 41 available, from each municipality in order to study a sample population as representative as possible of the families of Milan.”
The study design has also been clarified.
The validity and reliability of the instruments used are not considered, leaving the methods questionable.
ANSWER: the questionnaire used in the present study is a short form of the COHIP (Child Oral Health Impact Profile) that has already had its validity and reliability extensively examined in several studies.
- Broder, H.L.;Wilson-Genderson, M. Reliability and Convergent and Discriminant Validity of the Child Oral Health Impact Profile (COHIP Child's Version). Community Dent Oral Epidemiol 2007, 35(1), 20-31.
- El Osta N, Pichot H, Soulier-Peigue D, Hennequin M, Tubert-Jeannin S. Validation of the child oral health impact profile (COHIP) french questionnaire among 12 years-old children in New Caledonia. Health Qual Life Outcomes. 2015;13:176
- Agnew CM, Foster Page L, Hibbert S. Validity and reliability of the COHIP-SF in Australian children with orofacial cleft. Int J Paediatr Dent. 2017;27(6):574-582.
For this reason, we decided not to test the questionnaire again. The following sentence has been added in the text: “ . Part of the questions were aimed at evaluating the socio-emotional influence of oral health issues on the children and families and were adapted from the COHIP (Child Oral Health Impact Profile) of which the reliability and validity have been extensively measured.” A further reference has also been added to sustain the validity of the test.
Ethical considerations, informed consent of children and guardians are not mentioned.
ANSWER: The following sentences have been added: “The activity (survey) was explained to the parents at the meeting preceding the start of the summer camp. The families were informed on the goal of the study, told that the survey was meant to be filled in anonymously and that the data would be treated and analyzed by the Municipality of Milan and the University of Milan. The same information was reminded by means of an opening statement on the front page of the survey. By returning the survey, the consent was therefore granted. “
I would be hesitant, despite what seems like a rigorous analysis and application of statistical methods to assess the results in detail. also the discussion is also dependent on the framing of the introduction which would require considerable revision if the manuscript can be salvaged.
ANSWER: The manuscript has been deeply rewritten to make all the study more understandable. Introduction and discussion are now more adherent with the methods and results of the study. Also methods and results have been deeply modified.
Does the paper anything new or novel which we already do not know? This remains unexplained in the paper .
ANSWER: Data of the present study derive from a population as representative as possible of the families of Milan and are lacking in literature. Results will be used as the ground on which to design the following prevention programmes of oral health for pre-scholar and scholar children on Milan. This point has been added in conclusion.

Reviewer 3 Report
Impact of lifestyle variables on oral diseases and oral health related quality of life: a survey in children aged 6 to 11 years attending summer schools at Milan, Italy
This article has some methodological and style or presentation aspects to improve before acceptance for publication:
Abstract:
- Study design is not mentioned. The conclusion mentioned in the abstract is not directly related to the study results.
Introduction:
- The affirmation “On the other hand, the knowledge on the causes of oral diseases such as caries and periodontal inflammation and on how to prevent them among kids and their families, is often scarce” (page 2, lines 56-58) is not supported by any bibliographic reference that supports it. Does this statement also derive from 6th reference?
- In the introduction it is necessary to delve into one of the main outcomes such as the Oral health related quality of life. Oral health related quality of life is a widely studied construct that not only addresses the percentage of children who have experienced toothache or who have lost one or more days at school.
- Socio-economic variable is not mentioned in the (i) objective.
Materials and Methods:
- In section Study design type of study design is not described.
- The purpose of explaining educational intervention to children during school is not understood if its impact on the oral health of the children or on the knowledge of the parents afterwards is not measured (page 2, lines 85-90).
- In the statistical analysis section, it is not necessary to explain that “the surveys were collected, the data entered in an Excel sheet and analyzed in such a way to obtain the proportion of answers for each question” (page 3, lines 107-108).
Results:
- The way of mentioning the subsections of the Results section is not understood. This reviewer proposes the following subsections: 3.1 Description of the participants, 3.2 main outcome variables, 3. 3 other variables... It is not understood why a section 3.3 of data analysis has been specified.
- It does not make much sense to include the data in Table 1 as they are not collected after in the data analysis.
- On line 125 after 25% of the 8 years a comma should appear and not a point.
- To measure the oral health related quality of life, beyond the experience of toothaches or having lost one or more days at school, it should have been objectified with some appropriate validated scale to assess the perception of quality of life.
- It is recommended to present the data in tables 2-5 with means (M) and standard deviations (SD). It is also recommended to improve the aesthetic presentation of the tables 2-5.
- It would be advisable to present in this article the Supplementary material (page 10, lines 330-335), that helps to better visualize and relate all the data.
Discussion:
- The information in paragraphs 1 and 2 (page 8, lines 214-222) is repeated and should not appear again in the discussion.
- There is no discussion of the oral health related quality of life main outcome variable.
Conclusions:
- The conclusions drawn in this article do not derive from the main results. This section doesn’t provide readers a brief summary of the main conclusions based on the principal results.
- The results of the article do not yield enough data to state that “…health professionals like gynecologists, pediatricians, dentist and general practitioners do not seem to convey such information consistently and effectively” (page 10, lines 322-323).
- In the conclusions section it is the first time that it is mentioned that this article responds to a pilot project. This information should previously appear in the material and methods section together with the explanation of the study design.
References:
- In reference 8 the link is missing (page 11, line 361).
I hope that the suggested changes help to improve the quality of the article and that they are well received.
Kind regards,
Author Response
Dear reviewer, thank you for your review of the manuscript titled "
"Impact of lifestyle variables on oral diseases and oral health related quality of life: a survey in children aged 6 to 11 years attending summer schools at Milan, Italy".
Following your indications we made changes to the overall text. Here below, we report the detailed answers to the your suggestions. Main changes in the text are highlighted in green and the revisited manuscript has been attached to these notes.
We think that the quality of the manuscript is widely improved.
Thank you for your consideration of our work for publication on IJERPH.
REVIEWER 3
This article has some methodological and style or presentation aspects to improve before acceptance for publication:
Abstract:
- Study design is not mentioned. The conclusion mentioned in the abstract is not directly related to the study results. ANSWER: the study design has been mentioned as follows: “ This is an observational study and is part of a larger project….” And conclusion of the abstract has been modified as follows: “Concluding, protective behaviors and socio-economic status affect the oral disease and OHRQoL in children of Milan.”
Introduction:
- The affirmation “On the other hand, the knowledge on the causes of oral diseases such as caries and periodontal inflammation and on how to prevent them among kids and their families, is often scarce” (page 2, lines 56-58) is not supported by any bibliographic reference that supports it. Does this statement also derive from 6th reference? ANSWER: The reference has been added
- In the introduction it is necessary to delve into one of the main outcomes such as the Oral health related quality of life. Oral health related quality of life is a widely studied construct that not only addresses the percentage of children who have experienced toothache or who have lost one or more days at school. ANSWER: the introduction has been rewritten to delve into the topic of the Oral health related quality of life
- Socio-economic variable is not mentioned in the (i) objective. ANSWER: the socio-economic variable has been mentioned in the objective.
Materials and Methods:
- In section Study design type of study design is not described. ANSWER: the study design has been mentioned as follows: “ This is an observational study that rapresents part of a larger project….”. Furthermore, the study design has been described with more details.
- The purpose of explaining educational intervention to children during school is not understood if its impact on the oral health of the children or on the knowledge of the parents afterwards is not measured (page 2, lines 85-90). ANSWER: in materials and methods the study design has been better explained. The following part has been added: “This is an observational study that represents the pilot part of a larger project designed by the Municipality of Milan (Italy) for the promotion of the oral health in children…. In 2018, the Municipality of Milan organized a one-day educational program at the schools with the aim of providing oral health information to the schoolchildren of Milan. A team of one dentist and one dental hygienist explained how to preserve good oral health and prevent oral diseases to the children, in every individual school over the 2 weeks period. The intervention was conceived as a game in which the children were invited to play the roles of bacteria, sugar, candies, soft drinks, juices, water, toothbrush and fluoride. At the end of the lesson, the children were given stickers with simple rules on how to preserve oral health to take home. A leaflet targeted for their parents on the same topics was also delivered.”
No clinical examination of children was foreseen by the educational program. On the occasion of this one-day meeting, a questionnaire was created for the children’s caregivers, that included questions on different domains: oral diseases, OHRQoL, oral hygiene habits, food consumption, economic status, educational status. In the present observational study, data from these questionnaires were analyzed.”
- In the statistical analysis section, it is not necessary to explain that “the surveys were collected, the data entered in an Excel sheet and analyzed in such a way to obtain the proportion of answers for each question” (page 3, lines 107-108). ANSWER: The sentence has been removed.
Results:
The way of mentioning the subsections of the Results section is not understood. This reviewer proposes the following subsections: 3.1 Description of the participants, 3.2 main outcome variables, 3. 3 other variables... It is not understood why a section 3.3 of data analysis has been specified. ANSWER: To make the manuscript more understandable, the result section has been organized as follows: 3.1 Description of the participants, 3.2 Disease and OHRQoL, 3.3. Lifestyle-related variables, 3.3.1. Effects of variables on disease
- It does not make much sense to include the data in Table 1 as they are not collected after in the data analysis. ANSWER: table 1 has been removed has suggested.
- On line 125 after 25% of the 8 years a comma should appear and not a point. ANSWER: the comma has been added.
- To measure the oral health related quality of life, beyond the experience of toothaches or having lost one or more days at school, it should have been objectified with some appropriate validated scale to assess the perception of quality of life. ANSWER: the following sentence has been added to discuss this issue: Though it has been shown that children and caregiver ratings on OHRQoL can be in disagreement, with caregivers rating their children’s QoL lower than the children rated themselves in the present study the children were not interviewed concerning their own perception as regulations did not allow it [46].
- It is recommended to present the data in tables 2-5 with means (M) and standard deviations (SD). It is also recommended to improve the aesthetic presentation of the tables 2-5. ANSWER: the reported data are descriptive of the population and represent the % of answers for each question. Therefore, these data cannot be reported as mean values and standard deviations. The table format is indicated by the editor in the template for the manuscript submission and probably cannot be changed.
- It would be advisable to present in this article the Supplementary material (page 10, lines 330-335), that helps to better visualize and relate all the data. ANSWER: Since the material reported in the present manuscript is full-bodied and long, we decided to report part of data in supplementary material. If Editor deems it appropriate, the Supplementary material can be moved in the main text.
Discussion:
- The information in paragraphs 1 and 2 (page 8, lines 214-222) is repeated and should not appear again in the discussion. ANSWER: in the first part of discussion data were reported to facilitate the comparison with data reported in literature. Otherwise, the following sentence has been removed: “Concerning food habits, in the present study 7% of the children had received a pacifier dipped in sugar/honey, 29% had been given sweet tisanes in baby bottles, 47% ate snacks frequently, 58% drank often juices and soft drinks and 58% ate frequently candies and chocolate. Only about half of the families reported they had received information concerning how to preserve their children’s oral health by their family doctor/pediatrician.”
- There is no discussion of the oral health related quality of life main outcome variable. Answer: the discussion on the oral health related quality of life has been added
Conclusions:
- The conclusions drawn in this article do not derive from the main results. This section doesn’t provide readers a brief summary of the main conclusions based on the principal results. ANSWER: Conclusion has been rewritten to be more adherent with the results of the study.
- The results of the article do not yield enough data to state that “…health professionals like gynecologists, pediatricians, dentist and general practitioners do not seem to convey such information consistently and effectively” (page 10, lines 322-323). ANSWER: the sentence has been rephased.
- In the conclusions section it is the first time that it is mentioned that this article responds to a pilot project. This information should previously appear in the material and methods section together with the explanation of the study design. ANSWER: in materials and methods, the study design has been better explained. The following part has been added: “This is an observational study that represents the pilot part of a larger project designed by the Municipality of Milan (Italy) for the promotion of the oral health in children….”
- References: In reference 8 the link is missing (page 11, line 361). ANSWER; the link has been added.
I hope that the suggested changes help to improve the quality of the article and that they are well received.

Author Response
Dear reviewer, thank you for your review of the manuscript titled "
"Impact of lifestyle variables on oral diseases and oral health related quality of life: a survey in children aged 6 to 11 years attending summer schools at Milan, Italy".
Following your indications we made changes to the overall text. Here below, we report the detailed answers to your suggestions. Main changes in the text are highlighted in green, and the revited manuscpt has been attached to these notes.
We think that the quality of the manuscript is widely improved.
Thank you for your consideration of our work for publication on IJERPH.
REVIEWER 4
Title
- It is necessary to shorten the title of the study and reorganize. ANSWER: the following title is proposed: “Impact of lifestyle variables on oral diseases and oral health-related quality of life in children of Milan (Italy)”
- Authors need to organize keywords so that your keyword matches in the title. ANSWER: the keywords were modified: Children; Oral disease; Italy; Child; Lifestyle; oral disease; oral health-related quality of life; feeding habits; oral hygiene; socio-economic status
- Abstract - The first sentence of the abstract seems to briefly describe the background and necessity of the study. ANSWER: The first sentence of the abstract has been rephrased as follows: a large part of the Italian population doesn’t receive though life adequate information and support on how to preserve their oral health In the study, we investigated how some lifestyle-related variables affect the oral disease and oral health-related quality of life (OHRQoL)of children attended the public-school summer services in Milan were investigated.
- I wonder if a full name for ISEE is necessary in the abstract. ANSWER: “(equivalent family income)” has been added.
- Introduction
- In the first part of the introduction, we need to elaborate on the current state of oral health, especially the recent issues of child oral and family issues. (References should be inserted if necessary; eg, Some of their associated risk factors, like poor oral hygiene and elevated sugar intake, are easily modifiable, as well as known protective factors, such as the use of fluoride to prevent caries development( Ref ? ). ANSWER: the introduction has been rewritten to make it more adherent to the methods of the study and a part on the recent issues of child oral and family issues has been added. Several references have also been added.
- Briefly explain the necessity and importance of quality of life related to oral health in necessity. ANSWER: the introduction has been rewritten to delve into the topic of the Oral health related quality of life
Materials and Method - A detailed explanation of the research method is needed. ANSWER:Material and methods have been extensively to make them more clear and detailed.
- To make research progress more clearly and not to overlap. Analysis is repeated twice such as. In this part and result section..(Line 139, Data analysis). ANSWER: in this section, data analysis has not been repeated. Actually, it has been reported that some variable categories were aggregated due to numerosity issues to assess how these lifestyle-related variables affect the oral disease and OHRQoL. The following sentence has been modified to make this concept clearer: “To assess how these lifestyle-related variables affect the oral disease and OHRQoL, with respect to the original questionnaire, the following variable categories were aggregated due to numerosity issues:”
- How many records of all study participants? ANSWER: the following sentence has been added: A total of 478 surveys were delivered and 296 of these have returned
- What is the consent process before the questionnaire to family and the ethical problem solving process? ANSWER: The following sentences have been added: “The activity (survey) was explained to the parents at the meeting preceding the start of the summer camp The families were informed on the goal of the study, told that the survey was meant to be filled in anonymously and that the data would be treated and analyzed by the Municipality of Milan and the University of Milan. The same information was reminded by means of an opening statement on the front page of the survey. By returning the survey, the consent was therefore granted. “
- Data collection& Data analysis
- In the data collection and data analysis parts, please organize them to be more concise and understandable. ANSWER: Table 1 has been removed. The data collection and data analysis parts have been deeply modified to make them more understandable.
- Line 91-94 “Data collection. In the present study, two main outcomes were defined” Was it results? Or..
(I could not understand it. Sorry about that).
ANSWER: to make it more understandable the text was modified as follows:
“Data collection.
In the present study, the first part of the questionnaire evaluated the children’s:
- Disease: the current or past presence of tooth decay or tooth related abscess (yes/no);
- OHRQoL: current/past toothache or loss of school days due to toothache (yes/no).
The following lifestyle-related variables of children and their families were then investigated:
- Feeding habits….”
- Result - Overall, it would be better to remove unnecessary research results from the table and make the table more concise. (eg. Is the P value also italicized?): ANSWER: Table 1 has been removed, as suggested also by another review. P value has not been reported in tables since they only report descriptive data.
- Conclusion The conclusions need to be explained in more detail and include suggestions. ANSWER: conclusions have been modified and suggestions have been added. “Concluding, protective behaviors and socio-economic status affect the oral disease and OHRQoL in children of Milan. Such results suggest that families and caregivers could profit from a deeper knowledge on the most common oral diseases concerning protective behaviors that could help preventing their onset. Health professionals like gynecologists, pediatricians, dentist and general practitioners may collaborate in conveying such information consistently and effectively. Furthermore, data of the present study are the ground on which would be designed the following prevention programmes of oral health for pre-scholar and scholar children on Milan.”

Round 2
Reviewer 3 Report
Second revision of the article:
Impact of lifestyle variables on oral diseases and oral health-related quality of life in children of Milan (Italy)
The article continues presenting the following important methodological aspects to improve for its publication:
- The abstract continues without mentioning the study design.
- In the introduction, in the newly added reference number 15 the electronic link is missing. The reference is incomplete in the list of references.
- The conceptual framework still does not address the oral health-related quality of life (OHRQoL) construct. OHRQoL is an integral part of general health and well-being and a multidimensional construct that includes a subjective evaluation of the individual’s oral health, functional well-being, emotional well-being, etc. This construct is described very briefly and, in this study, it is not measured with any validated scale. The present study included questions concerning OHRQoL such as experiencing toothache or missing school days, but it is not clear how the OHRQoL has been measured.
- As the authors explain, it is a very important limit of this study that no oral examination was performed on the children and therefore the answers concerning disease prevalence could not clinically confirmed.
- The part of the discussion on the main variable OHRQoL is insufficient.
- Although the conclusions have been rewritten, through the assessment of parental perception it cannot be concluded that protective behaviors and socio-economic status affect the oral disease and OHRQoL in children of Milan when no oral examination was performed and when OHRQoL has not actually been measured with a validated scale that groups all the dimensions contained in this multidimensional construct.
I hope that the suggested changes help to improve the quality of the article and that they are well received.
Kind regards